# Relationship between Toothpaste Dilution Ratio and Droplets Generated during Tooth-Brushing

**DOI:** 10.3390/ijerph19074157

**Published:** 2022-03-31

**Authors:** Ryouichi Satou, Atsushi Yamagishi, Atsushi Takayanagi, Takuro Higuchi, Tsutomu Oyama, Seitaro Suzuki, Naoki Sugihara

**Affiliations:** 1Department of Epidemiology and Public Health, Tokyo Dental College, Tokyo 101-0061, Japan; yamagishiatsu@tdc.ac.jp (A.Y.); takayanagi@tdc.ac.jp (A.T.); suzukiseitarou@tdc.ac.jp (S.S.); sugihara@tdc.ac.jp (N.S.); 2Personal Health Care Products Research, Kao Corporation, Tokyo 131-0044, Japan; ooyama.tsutomu@kao.com; 3Sensory Sciense Research, Kao Corporation, Tokyo 131-0044, Japan; higuchi.takuro@kao.com

**Keywords:** coronavirus, droplet, tooth-brushing, toothpaste

## Abstract

Guidelines for using toothpaste during tooth-brushing in public places during the coronavirus epidemic are lacking. In addition, the advantages and disadvantages of using toothpaste in terms of droplet generation during brushing, the number of droplets generated, and their scatter range are unknown; therefore, we investigated the relationships between diluted toothpaste viscosity, the number of droplets generated, and the droplets’ flight distance. We developed a system to quantitate droplet generation during tooth-brushing. Brushing with water generated 5965 ± 266 droplets; 10.0× diluted toothpaste generated 538 ± 56, 4.00× diluted toothpaste generated 349 ± 15, and 2.00× diluted toothpaste generated 69 ± 27 droplets. Undiluted toothpaste generated no droplets. Droplet number tended to increase with increased toothpaste dilution ratio and decreased viscosity (*r* = −0.993). The maximum flight distances were 429 ± 11, 445 ± 65, 316 ± 38, and 231 ± 21 mm for water, 10.0×, 4.00×, and 2.00× diluted toothpaste, respectively. The maximum flight distance and toothpaste viscosity correlated negatively (*r* = −0.999). Thus, the less diluted the toothpaste, the fewer the droplets generated during brushing, and the shorter their flight distance. The use of an appropriate amount of toothpaste is recommended to prevent droplet infection during tooth-brushing.

## 1. Introduction

The new coronavirus disease 2019 (COVID-19) is a worldwide pandemic [1,2]. Severe acute respiratory syndrome coronavirus 2 (SARS-CoV-2), which causes COVID-19, has been reported to have an attachment pattern in the human body similar to that of the influenza virus [3]. The virus adheres to airway epithelial cells via the hemagglutinin (HA) glycoprotein, which protrudes from the viral envelope surface, and allows viral entry into cells [4]. Viral infections are associated with a large amount of virus adhering to the mucosa. Appropriate oral cleaning has been reported to reduce the viral load in the oral and pharyngeal mucosa, which suggests the possibility of preventing viral infection by cleaning the oral cavity [4,5]. The oral cavity is the entrance to the lower respiratory tract, such as the bronchi and lungs, where inflammation is induced upon SARS-CoV-2 infection, and the back of the tongue, gums, and salivary glands are growth sites for this virus [6,7]. Additionally, angiotensin-converting enzyme II (ACE2), a host protein that binds to the new coronavirus, and transmembrane protease serine 2 (TMPRSS2), which promotes infection, are expressed in the oral mucosal epithelium [8]. It has been reported that strengthening oral immunity helps prevent infection [8]. Moreover, multiple components of toothpaste, such as sodium tetradecene sulfonate, sodium lauroylmethyl taurine, sodium lauryl sulfate, and cetylpyridinium chloride (CPC), inhibit or inactivate the invasion of the new coronavirus into the body. It has been suggested that using toothpaste during tooth-brushing may thus be useful in preventing infection [4,8]; however, recommendations for the use of toothpaste are not specified in instructions for tooth-brushing in public places, such as companies and schools, during the coronavirus epidemic [9,10,11].

SARS-CoV-2 is found in the saliva of infected patients, and it has been reported that saliva droplets during tooth-brushing can be a source of infection [7]. Droplets of saliva are observed during brushing, and it is predicted that a large number of droplets will be generated if the toothbrush is moved to scrape the palatal side of the maxillary anterior teeth or if brushing is performed without closing the mouth [7]. When brushing teeth in public places, such as schools, sufficient ventilation, avoiding conversation, and discharging from a low position with a small amount of water should be encouraged [5,9]; however, the advantages and disadvantages of using toothpaste and the effects of droplets generated during brushing have not yet been investigated. In the COVID-19 pandemic context, this is also of high importance to consider when addressing public health. It is necessary to study the number of droplets generated and the range of scattering by brushing with toothpaste.

In previous research, we investigated changes in physical properties, such as viscosity and flow, resulting from the dilution of toothpaste [12]. It has been reported that the viscosity of toothpaste decreases sharply due to dilution and that the viscosity of the solution is related to the generation of droplets [13]. We hypothesized that changes in viscosity due to dilution affect the number of droplets generated and the flight distance of these droplets.

Thus, this study aimed to clarify the relationship between the viscosity of diluted toothpaste, the number of droplets generated, and their flight distance. We also developed an evaluation system that can quantitatively measure droplet generation during tooth-brushing.

## 2. Materials and Methods

### 2.1. Study Design and Development of a Droplet Evaluation Model Using a Single-Axis Robot

We aimed to develop an evaluation system that can quantitatively measure droplet generation during tooth-brushing as an experimental study. In this study, we developed a device that moved a toothbrush at a constant velocity. We attached a toothbrush head via a custom-made arm to a single-axis robot (RS1, Misumi, Tokyo, Japan) and created a brushing robot that moved linearly at a constant velocity (Figure 1). A droplet generation unit with grooves carved at equal intervals was installed at the bottom of the arm of the brushing robot. A certain amount of strain was placed on the toothbrush head by the grooves, and when it overcame the grooves, the strain was released, and droplets were generated. The droplet generation unit reproduced the palatal side of the maxillary anterior teeth, and by changing the height of the step at the tip, the amount of deflection of the toothbrush head was adjusted by 2 mm. A number of grooves were used in the droplet generation unit to remove excess liquid and to make the amount of toothpaste adhere to the brush uniformly. The brushing speed was set to 200 mm/s based on previous research. The structure of the toothbrush head (Original prototype for this research (not for sale), KAO Corp., Tokyo, Japan) was simplified. It was made of nylon, with a diameter of 8 mil (200 μm), length of 11 mm, tuft diameter of 1.6 μm, and flocking pattern of 4 tufts per row. The droplets created by the robot were dropped onto inkjet paper (KA4100SFR, Super Fine paper, A4, Seiko Epson Corp.,Tokyo, Japan, measurement range: W210 × L594 mm, connect two A4 sheets) and recorded. The measurement sample was printed on an inkjet by generating splashes by reciprocating motion 5 times. Measurements were performed 5 times for each dilution. The inkjet paper was changed after each measurement. The inkjet paper was digitized with a scanner, and the number of droplets, flight distance, and droplet size were calculated using image analysis software (ImageJ 1.52 p, NIH, Bethesda, MD). The flight distance was measured with the incisal edge of the droplet generation unit being 0 mm. The linear distance between the point where the droplets were generated and the point where the inkjet paper fell was measured as the flight distance.

### 2.2. Preparation of a Model Toothpaste

In this study, we created a model toothpaste that reproduces the components of common commercially available toothpaste [14]. The composition of the model toothpaste was distilled water 34.5%, sorbitol 30.0%, calcium carbonate (Calcium # 9860, TCI, Tokyo, Japan), 15%, glycerin 10.0%, thickening silica (SLP25) 6.0%, carboxymethyl cellulose (CMC, F35SH, Wako, Tokyo, Japan), 2.4%, lauryl sulfate (1614363, Sigma-Aldrich, St. Louis, MO, USA), 1.2% sodium, 0.8% fragrance, and 0.1% sodium saccharin (47839, Sigma-Aldrich, St. Louis, MO, USA). The toothpaste was diluted with distilled water to prepare five groups, with concentrations of 100% (1.00×), 67% (1.50×), 50% (2.00×), 25% (4.00×), and 10% (10.0×). 4.00× foams well, and in the foam state, the behavior at the time of droplet generation is different; therefore, it was divided as 4.00× (Form). Additionally, water without toothpaste was used for comparison. Red No. 102 (1.0%) was added when the toothpaste was diluted to facilitate visualization of the droplets.

### 2.3. Viscosity of Toothpaste

A sinewave vibro (tuning fork-type vibration type) viscometer (SV-10, A&D Corp., Tokyo, Japan) was used to measure the viscosity of the toothpaste. The measurement was conducted at a temperature of 25 °C, according to the manufacturer’s instructions.

### 2.4. High-Speed Camera Observation at the Time of Droplet Generation

The moment of droplet generation was photographed with a high-speed camera (Photron FASTCAM SA3 Model 120 k, Photon Ltd., Tokyo, Japan) and a macro lens (Ai AF Micro Nikkor 105 mm F2.8 D; Nikon Corp., Tokyo, Japan). Photographs were taken at a frame rate of 6000 fps, a shutter speed of 1/6000 s, a lens aperture of f8, and a shooting size of 521 × 512 pixels.

### 2.5. Statistical Analysis

Data for each group are presented as the mean ± standard deviation of five replicates. In the experiment on droplet amount and maximum flight distance, groups were compared by one-way analysis of variance (ANOVA), and differences were considered significant at *p* < 0.01. The Bonferroni test was used for post hoc comparisons when this analysis yielded *p* < 0.01 (Origin Pro 2022, OriginLab Corp., Northampton, MA, USA). A scatter plot was created to clarify the relationship between the concentration and viscosity of toothpaste, the number of droplets generated. The correlation coefficient was calculated from the approximate expression.

## 3. Results

### 3.1. Viscosity of Toothpaste According to Dilution Ratio

Table 1 and Figure 2 show the changes in toothpaste dilution ratio and viscosity. At a concentration of 100% (1.00×), i.e., undiluted toothpaste, the toothpaste viscosity was very high. When the toothpaste was diluted to a concentration of 67% (1.50×), the viscosity dropped sharply to approximately 1/10th of that of the 100% (1.00×) sample. The viscosity further decreased as the concentration decreased (Table 1). The amount of decrease of 100% (1.00×) and 67% (1.50×) was the largest; after 1.5×, the decrease became more gradual.

Figure 2 provides a scatter plot showing the correlation between the dilution ratio of toothpaste and its viscosity (Figure 2). A strong positive correlation was found between the toothpaste concentration and viscosity (*r* = 0.999).

### 3.2. Amount and Properties of Droplets for Each Toothpaste Dilution Ratio

Figure 3 shows an inkjet-paper image recording the droplet properties for each toothpaste dilution ratio. The number of droplets was highest in water and tended to decrease when the dilution ratio was low. The number of droplets and the droplet size were both large in water, decreased sharply between water and the 10.0× dilution of toothpaste. More fine droplets were observed in water than when toothpaste was present. Table 1 shows the viscosity, number of droplets, and maximum flight distance of the droplets according to the dilution ratio of the toothpaste. The number of droplets generated tended to increase as the dilution ratio of the toothpaste increased and the viscosity decreased. In the case of 1.00× samples, no droplets were observed (Table 1). Figure 4 shows a comparison of the number of droplets generated at each toothpaste dilution ratio. The number of droplets was greatest for water, then reduced to almost 1/10th of this value for the 10× dilution, and thereafter reduced more gradually. Water and 1.00× samples showed statistically significant differences from all other dilutions (*p* < 0.01) (Figure 4). Statistically significant differences were also observed between 10.0× and 4.00× (foam) and between 10.0× and 2.00× (both *p* < 0.01). At 4.00×, there was a difference in the dynamics of droplet generation depending on the presence or absence of foaming created by the brushing motion; therefore, we distinguished this as 4.00× (foam). A scatter plot was created to clarify the relationship between the concentration and the number of droplets generated (Figure 4). There was a strong negative correlation between the number of droplets generated and the viscosity of the toothpaste (*r* = −0.993).

### 3.3. Maximum Droplet Flight Distance for Each Toothpaste Dilution Ratio

Table 1 and Figure 4 show a comparison of the maximum flight distances for each toothpaste dilution ratio. The maximum flight distances were 429 ± 11, 445 ± 65, 316 ± 38, and 231 ± 21 mm for the water, 10.0×, 4.00×, and 2.00×, respectively. In the case of 1.00×, no droplets were observed (Figure 4). At 4.00 B×, the number of droplets when foaming occurred was 183 ± 49, showing a decreasing tendency as compared to when not foaming, but the difference was not statistically significant (*p* > 0.01). A scatter plot was created to clarify the relationship between the dilution concentration and maximum droplet distance (Figure 4). There was a strong negative correlation between the maximum flight distance and toothpaste viscosity (*r* = −0.999).

### 3.4. Changes in the Number of Droplets and Flight Distance for Each Toothpaste Dilution Ratio

Figure 5 shows the transition of the number of droplets within the flight distance range based on the toothpaste dilution ratio. At all dilution concentrations except 1.00×, the number of droplets within the range tended to decrease as the distance from the droplet generating portion increased.

Only water and the 10.0× samples showed droplets beyond 400 mm (Figure 5). The number of droplets decreased sharply from 150 mm for all dilution ratios, and the number of droplets was fewer than 100 within a range of 150 mm or more, except for water. Comparing 4.00× with and without foaming, the number of droplets generated was reduced when foaming was present.

### 3.5. High-Speed Camera Observation at the Moment of Droplets Generation

Figure 6 shows an image of the moment when droplets were generated as captured with a high-speed camera. In the case of water, a large number of water droplets were generated towards the front, lower region at the moment when the deflection of the hair bundle was released, and fine droplets were scattered (Figure 6). For 10.0× samples, the toothpaste adhered to the brush for a long time after the deflection had been released, and scattered forward and diagonally upward (Figure 6). For 4.00× samples, the number of water droplets generated was further reduced as compared to 10.0× samples (Figure 6). At 4.00×, the dynamics differed with and without foam. The number of droplets flying forward was suppressed in the case of foaming, as compared to the case of no foaming, and the entire foam vibrated, while no fine droplets were generated (Figure 6). Owing to the high viscosities of 2.00× and 1.00×, the toothpaste was seen to cling to the brush when the deflection was released, and almost no droplets were observed (Figure 6).

## 4. Discussion

This study clarified that with less diluted toothpaste, a smaller number of droplets are generated during tooth-brushing, with a shorter flight distance. Brushing with water alone creates the largest number of droplets and the largest flight distance, implying that the risk of infection due to droplets is relatively high. When about 1.0 g of toothpaste is used, it is diluted to 4.00× in the oral cavity [15]. If the amount of toothpaste used is less than 1.0 g, the dilution factor in the oral cavity will be greater than 4.00×. Many droplets were generated with the 4.00× sample, but when there was sufficient foaming, even at this dilution ratio, the generation of droplets and the flight distance were suppressed. Thus, to suppress the generation of droplets, it may be effective to use more than 1.0 g of toothpaste and to ensure that sufficient foam is created. Our data suggested that, in order to prevent COVID-19 infection during tooth-brushing, it is desirable that the toothpaste is not diluted by 4.00× or more, and that increasing the viscosity of the oral fluid could reduce the generation of droplets.

The evaluation system we created controlled each element related to the generation of droplets during brushing and had high reproducibility. Because this evaluation system was controlled by a robot, the contribution of each element, such as brushing speed, brushing pressure, toothpaste amount, and toothpaste dilution concentration, could be quantitatively isolated and evaluated. It is also possible to change the head of the toothbrush and the form of the chemical solution, and thus this system can be applied to various measurements related to brushing, not only the generation of droplets; however, this evaluation system does not directly measure the droplet size and cannot detect floating droplets that do not fall; therefore, future studies on aerosol visualization technology, such as droplet image velocimetry, are required.

The viscosity of the model toothpaste used in this study was adjusted with CMC and showed the same tendency as that of general water-soluble polymers. The relationship between dilution and viscosity may differ depending on the toothpaste components; however, it may be possible to estimate the effect on droplet generation by examining the dilution–viscosity relationship.

The effect of the presence or absence of toothpaste on droplets has not been investigated previously, and guidelines for brushing instruction in consideration of infection prevention do not indicate the use of toothpaste [3,11]. The results of our study suggest that the use of toothpaste during brushing should be recommended to control droplet generation. Several reports recommend the use of oral care products, such as toothpaste and mouthwash, for the reduction in salivary-borne viruses and the risk of infection. The Center for Disease Control and American Dental Hygienists’ Association recommends the use of oral care products containing CPC or chlorhexidine gluconate (CHX) to reduce viruses in the saliva. According to a report by Komine et al., oral care solutions containing CPC, CHX, and delmopinol hydrochloride inactivated the virus in vitro [5,16,17]. In addition, sodium tetradecene sulfonate, sodium N-lauroyl-N-methyltaurate, sodium N-lauroylsarcosinate, sodium dodecyl sulfate, and copper gluconate have been shown to inhibit the interaction between the SARS-CoV-2 spike protein and ACE2 and the protease activity of TMPRSS2 [18]. Nevertheless, these data are limited to in vitro studies and need to be evaluated in vivo; however, taken together with our findings, these results indicate that the combined use of a toothpaste containing an antibacterial component can reduce the amount of virus in saliva, suppress the generation of droplets, and reduce the risk of droplet infection, as compared to brushing with water alone. Our results indicate that, by reducing the dilution of the toothpaste, the concentration of fluoride and the above-mentioned antiviral components could be kept high, without requiring a change in the amount of toothpaste. It is recommended to use more than 1.0 g of fluoride-containing toothpaste for caries prevention, but it may be preferable to use a larger amount to prevent droplet generation [19].

## 5. Conclusions

This study showed that the less toothpaste is diluted, the fewer droplets are generated during tooth-brushing, and the shorter the flight distance of the droplets. It has been suggested that the use of a large amount of toothpaste during brushing can suppress droplet generation. The use of an appropriate amount of toothpaste is recommended to prevent droplet infection during tooth-brushing.

## Figures and Tables

**Figure 1 ijerph-19-04157-f001:**
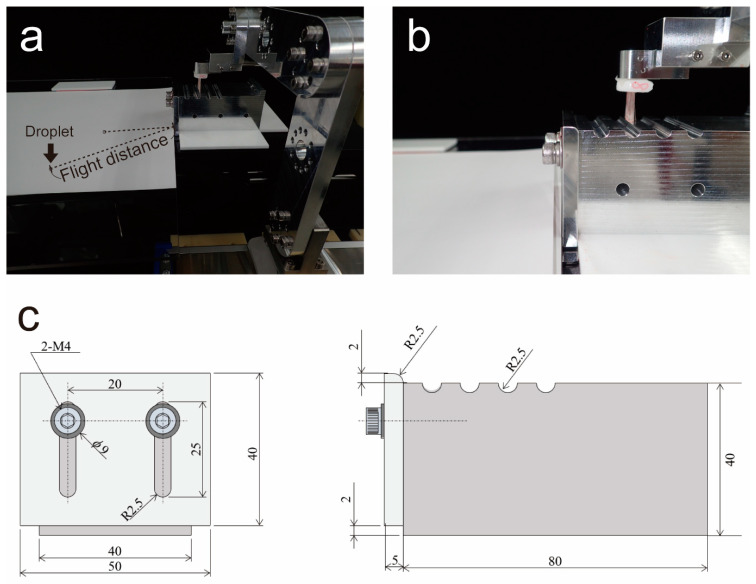
Overview of the droplet evaluation model using a single-axis robot. (**a**) Photograph of the uniaxial robot. A toothbrush head was attached to the arm, which repeated linear motion at constant velocity. The droplet generation unit at the bottom of the arm caused the brush to bend and droplet droplets generated were recorded on A4 paper. The linear distance between the droplet’s generated point and the droplet fell point was measured as the flight distance; (**b**) an image of the droplet generation unit is presented. The deflection of the toothbrush head could be adjusted by reproducing the palatal side of the maxillary anterior teeth and changing the height of the step at the tip. Using a number of grooves in the droplet generation unit allows removal of excess liquid and ensures that the amount of toothpaste adheres to the brush uniformly; (**c**) schematic diagram of the droplet generation unit. The distance is shown in mm. ϕ mean diameter; 2-M4 means to use two 4 mm diameter screws.

**Figure 2 ijerph-19-04157-f002:**
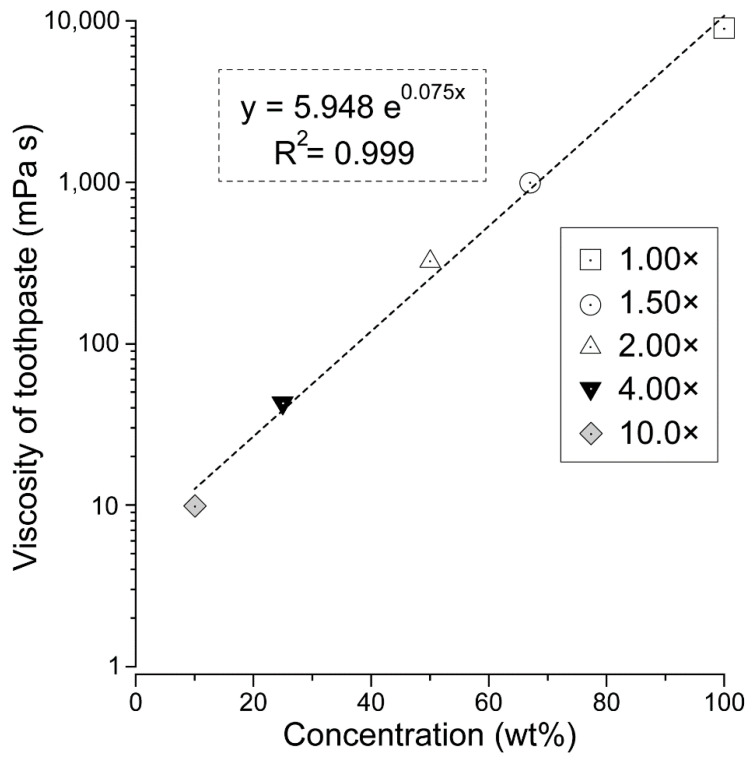
Relationship between toothpaste concentration and viscosity. The viscosity and correlation of each toothpaste concentration (1.00×, 1.50×, 2.00×, 4.00×, and 10.0×) are shown. The white square indicated dentifrice with a concentration of 100% (1.00×). White circles: 67% (1.50×); white triangles: 50% (2.00×); black inverted triangles: 25% (4.00×); gray diamonds: 10% (10.0×). A scatter plot was created with the logarithmic value of viscosity on the vertical axis and the concentration (%) on the horizontal axis. It was found that there was a strong positive correlation between toothpaste concentration and viscosity (*r* = 0.999).

**Figure 3 ijerph-19-04157-f003:**
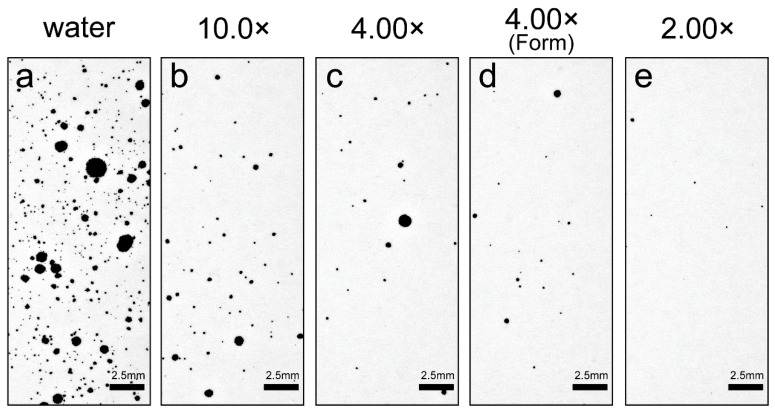
Comparison of droplet properties according to the toothpaste dilution ratio. (**a**–**e**) Properties of droplet droplets recorded at each toothpaste dilution ratio are shown. The scale bar represents 2.5 mm. It was created from an image in the range of 30–40 mm from the droplet generation points on A4-sized inkjet-recording paper. We chose this range setting as within the 0–30 mm range, there are many liquid components, the shape of the droplets is unstable, whereas the droplet properties are stable, and the characteristics of each dilution concentration are clearly represented at 30–40 mm.

**Figure 4 ijerph-19-04157-f004:**
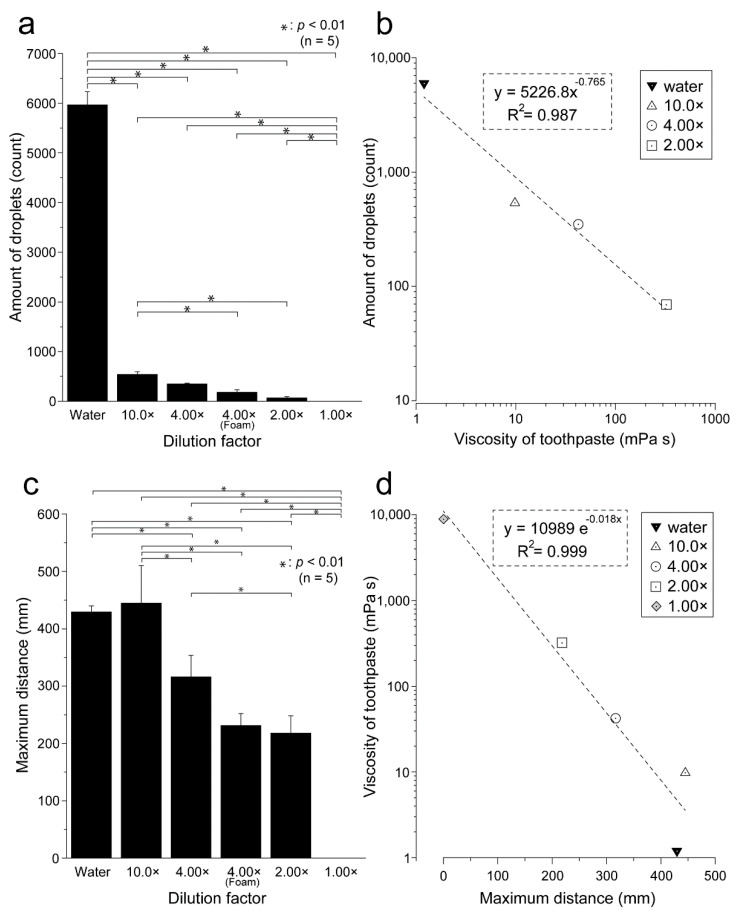
Comparison of the number of droplets generated and the maximum droplet distance by the toothpaste dilution ratio. (**a**) The relationship between the toothpaste dilution ratio and the number of droplets is shown. The number of droplets is presented as the mean ± standard deviation of five replicates per dilution ratio (*n* = 5); (**b**) the concentration of each toothpaste dilution (2.00×, 4.00×, 10.0×), the viscosity of water, and the number of droplets is shown. The black inverted triangles indicated dentifrice with water. White triangles: 10% (10.0×); white circles: 25% (4.00×); white square: 50% (2.00×); gray diamonds: 100% (1.00×). A scatter plot was created with the logarithmic value of the number of droplets generated on the vertical axis and the logarithmic value of the viscosity on the horizontal axis. There was a strong negative correlation between toothpaste viscosity and the number of droplets generated (*r* = −0.993); (**c**) the relationship between the toothpaste dilution ratio and the maximum flight distance is shown; (**d**) there was a strong negative correlation between toothpaste viscosity and the number of droplets generated (*r* = −0.999).

**Figure 5 ijerph-19-04157-f005:**
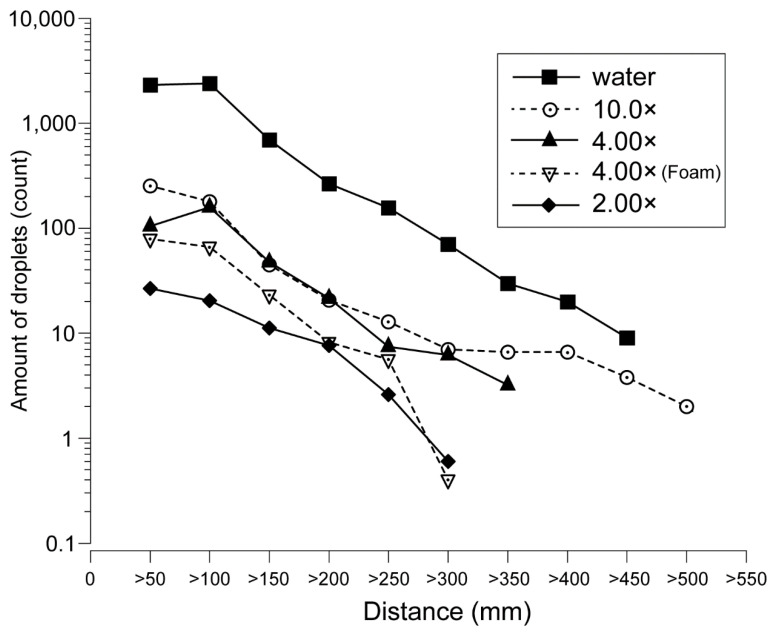
Comparison of the number of droplets within the flight distance range for each toothpaste dilution ratio. The black square indicated dentifrice with water. White circles: 10% (10.0×); black triangles: 25% (4.00×); white inverted triangles: 25% (4.00×, form); black diamonds: 50% (2.00×). The logarithmic value of the number of droplets is shown on the vertical axis and the flight distance range is shown on the horizontal axis. At all dilution concentrations except 1.00×, the number of droplets within the range tended to decrease as the distance from the droplet generating portion increased.

**Figure 6 ijerph-19-04157-f006:**
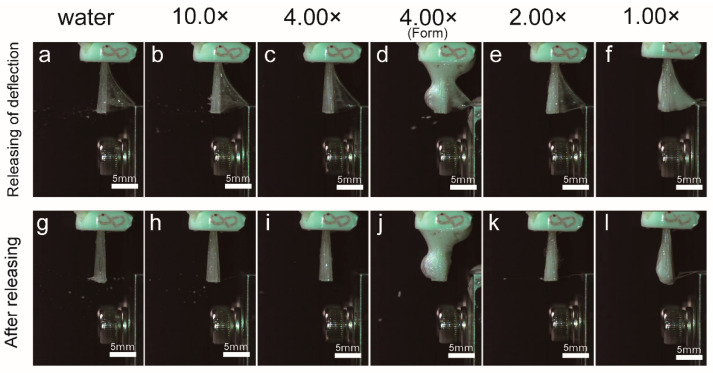
High-speed camera observation at the moment of droplet generation. Images taken with a high-speed camera at the moment and immediately after the droplet formation occurred are shown. (**a**–**f**) The moment when droplets are generated. (**g**–**l**) Immediately after the droplets are generated, deflection is released. Scale bars represent 5 mm.

**Table 1 ijerph-19-04157-t001:** Comparison of viscosity, droplet amount, and maximum droplet distance by toothpaste dilution ratio.

	Concentration (wt%)	Viscosity (mPa × s)	Droplets Count	Maximum Distance (mm)
Water	0	1.2	5965 ± 266	429 ± 11
10.0×	10	9.8	538 ± 56	445 ± 65
4.00×	25	42.5	349 ± 15	316 ± 38
4.00× (Form)	-	-	183 ± 49	231 ± 21
2.00×	50	324	69 ± 27	218 ± 30
1.00×	100	8930	0	0

## Data Availability

The data that support the findings of this study are available from the corresponding author, RS, upon reasonable request.

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
