# Peer review of "Relationship between Toothpaste Dilution Ratio and Droplets Generated during Tooth-Brushing"

_ijerph, 2022, doi:10.3390/ijerph19074157_

Round 1

Reviewer 1 Report

General comments

  • Keep full forms of ACE2 and
  • Use past tense in Methods section.

 Specific comments

Materials and Methods: Mention study design, sample size, sampling technique and data collection methods employed.

Results: Please clarify whether this was positive or negative correlation in the sentence in lines 139-140.

 Discussion

  • Lines 265-267: Keep citation and its reference to support the statement.
  • Lines 273-275: Rewrite the sentence clearly.

 References: Rewrite reference 2 clearly.

Author Response

Response to Reviewer 1 Comments

Thank you very much for providing important comments. We are thankful for the time and energy you expended. Our responses to the referees’ comments are as follow:

<General comments>

Keep full forms of ACE2 and Use past tense in Methods section.

> We appreciate the reviewer's comment on this point. In accordance with the reviewer's comment, we have changed this to following sentence:

Page 1, Line 39-40

Additionally, angiotensin converting enzyme II (ACE2), a host protein that binds to the new coronavirus,

We modified Method section to the past tense.

<Specific comments>

Materials and Methods: Mention study design, sample size, sampling technique and data collection methods employed.

> In accordance with the reviewer's comment, we have changed this to following sentence:

Page 2, Line 71-73

2.1. Study design and development of a droplet evaluation model using a single-axis robot

We aimed to develop an evaluation system that can quantitatively measure droplet genera-tion during tooth-brushing as experimental study.

Page 4, Line 136

Data for each group are presented as the mean ± standard deviation of five replicates.

Results: Please clarify whether this was positive or negative correlation in the sentence in lines 139-140.

> We regret this spelling mistake.

Page 4, Line 154-155

A strong positive correlation was found between the toothpaste concentration and viscosity (r = 0.999).

Discussion:

Lines 265-267: Keep citation and its reference to support the statement.

>Page 10, Line 291

We added references [3,11] to the sentence.

Lines 273-275: Rewrite the sentence clearly.

>The reviewer's comment is correct. We have changed this to following sentence:

Page 10, Line 297-298

According to a report by Komine et al., oral care solutions containing CPC, CHX, and delmopinol hydrochloride inactivated the virus in vitro.

References: Rewrite reference 2 clearly.

> In accordance with the reviewer's comment, we have changed reference to following sentence:

2.Zhonghua Liu Xing Bing Xue Za Zhi Zhonghua Liuxingbingxue Zazhi. The epidemiological characteristics of an outbreak of 2019 novel coronavirus diseases (COVID-19) in China, Epidemiology Working Group for NCIP Epidemic Response. 2020, 41, 145–151. DOI: 10.3760/cma.j.issn.0254-6450.2020.02.003.

Reviewer 2 Report

An English language review must be carried out. In addition, there are some grammatical errors and non-formal terms present in the text.

ABSTRACT

  • Lines 11, 12, 13, and 14: I suggest you rewrite the sentences for:

Guidelines for using toothpaste during tooth-brushing in public places during the coronavirus epidemic are lacking. In addition, the advantages and disadvantages of using toothpaste in terms of droplet generation during brushing, the number of droplets generated, and their scatter range are unknown. Therefore, we investigated the relationships of diluted toothpaste viscosity, the number of droplets generated, and the droplets’ flight distance.

KEYWORDS

I suggest replacing the keyword DROPLET and FLIGHT DISTANCE with MeSH terms (https://www.ncbi.nlm.nih.gov/mesh/).

Ex: Coronavirus

INTRODUCTION

  • Line 36

I suggest first writing the full name of SARS-CoV-2 and then putting the acronym in parentheses.

Author Response

An English language review must be carried out. In addition, there are some grammatical errors and non-formal terms present in the text.

> Thank you very much for providing important comments. We are thankful for the time and energy you expended. We asked a native speaker of Editage (www.editage.jp) to correct the English grammar.

Our responses to the referees’ comments are as follow:

ABSTRACT

Lines 11, 12, 13, and 14: I suggest you rewrite the sentences for:

Guidelines for using toothpaste during tooth-brushing in public places during the coronavirus epidemic are lacking. In addition, the advantages and disadvantages of using toothpaste in terms of droplet generation during brushing, the number of droplets generated, and their scatter range are unknown. Therefore, we investigated the relationships of diluted toothpaste viscosity, the number of droplets generated, and the droplets’ flight distance.

> Thank you for refining the text. This is very good. We agree with your assessment. we have changed sentence.

KEYWORDS

I suggest replacing the keyword DROPLET and FLIGHT DISTANCE with MeSH terms (https://www.ncbi.nlm.nih.gov/mesh/).

Ex: Coronavirus

> The reviewer's comment is correct. We have changed keywords.

INTRODUCTION

Line 36

I suggest first writing the full name of SARS-CoV-2 and then putting the acronym in parentheses.

> In accordance with the reviewer's comment, we have changed reference to following sentence:

Page 1, Line 28-30

Severe acute respiratory syndrome coronavirus 2 (SARS-CoV-2), which causes COVID-19, has been reported to have an attachment pattern in the human body similar to that of influenza virus [3].

Reviewer 3 Report

The manuscript made by Satou R et al., is interesting, well done, and provides relevant data regarding droplets in different density of toothpaste, using simulated models to evaluate several variables related to viscocity of toothpaste. Despite this is an interesting manuscript, I have some questions that the authors need to resolve.

1. Material and methods Lines 85 to 87. "After five..." I consider that these lines are confusing, and need to be better explained. I have some questions about it: 1. "After five..." did the repeated test measure the same inkjet paper? Or was it changed in each test? 2. "Flight distance..." I do not understand how the authors measured the flight distance of droplets, would it be possible to explain this better? 3. "Image analysis..." Despite the ImageJ is a good image analysis software, this needs several plug-ins to work properly. Would it be possible to explain whether a plug-in or plug-ins were used to measure droplets?   Results Figure 1, c) This figure does not properly represent the measure of flight distance of droplets. Would it be possible to provide a better scheme?   Table 1. Would it be possible to explain the difference between x4.00 and x4.00B? Within the text (material and methods) this was not explained. Moreover, why was not the 67% (1.5X) of concentration used in table 1?   Lines 150 to 152. "The number of droplets..." In this paragraph the authors describe the variables related to droplets and those that were put in table 1, however, why did not specify the size of droplets in the table 1, if the scale of the size of the droplets was specified in the material and methods section?   Line 293 to 294. “The use of an…” Would it be possible to describe how many toothpaste is recommended according to their results?   Lines 283 to 287. "Our results..." This paragraph is interesting, however, is confusing in the lines 293 to 294 "The use of an..." in which the ammount of toothpaste is described, would it be possible to review this paragraph and redact it in better manner?  

Author Response

The manuscript made by Satou R et al., is interesting, well done, and provides relevant data regarding droplets in different density of toothpaste, using simulated models to evaluate several variables related to viscocity of toothpaste. Despite this is an interesting manuscript, I have some questions that the authors need to resolve.

> Thank you very much for providing important comments. We are thankful for the time and energy you expended. Our responses to the referees’ comments are as follow:

  1. Material and methods Lines 85 to 87. "After five..." I consider that these lines are confusing, and need to be better explained. I have some questions about it:

> We have added following sentence to this section.

Page 2, Line 90-92

The measurement sample was printed on an inkjet by generating splashes by reciprocating motion 5 times. Measurements were performed 5 times for each dilution (n=5). The inkjet paper was changed after each measurement.

  1. "Flight distance..." I do not understand how the authors measured the flight distance of droplets, would it be possible to explain this better?

> We appreciate the reviewer's comment on this point. We have added following sentence to this section.

Page 3, Line 96-97

The linear distance between the point where the droplets were generated and the point where the ink jet paper fell was measured as the flight distance.

  1. "Image analysis..." Despite the ImageJ is a good image analysis software, this needs several plug-ins to work properly. Would it be possible to explain whether a plug-in or plug-ins were used to measure droplets?

> We are not using any additional plugins.

What we paid attention to when measuring was the quality of the image. Therefore, we have set conditions (dye density, paper selection, lighting, and other shooting conditions) that give good contrast to inkjet paper. 

Results

Figure 1, c) This figure does not properly represent the measure of flight distance of droplets. Would it be possible to provide a better scheme?  

> We appreciate the reviewer's comment on this point. We have added a diagram and explanation of how to determine the flight distance to Figure 1a.

Page 3, Line 102-103

The linear distance between the droplets generated point and the droplet fell point was measured as the flight distance.

Table 1. Would it be possible to explain the difference between x4.00 and x4.00B? Within the text (material and methods) this was not explained. Moreover, why was not the 67% (1.5X) of concentration used in table 1?  

> 4.00B means bubbling 4.00x. It has been corrected to 4.00 (Form).

The 1.50x sample was added only during the viscosity measurement to see the viscosity reduction behavior precisely. In the viscosity experiment, 1.50x was measured, but in this experiment of flight distance measurement, only 2.00x or later was measured. We have excluded items from Table 1 because we do not have 1.50x flight distance data.

We have added following sentence to Method section.

Page 3, Line 118-119

4.00× foams well, and in the foam state, the behavior at the time of droplet generation is different. Therefore, it was divided as 4.00× (Form).

Lines 150 to 152. "The number of droplets..." In this paragraph the authors describe the variables related to droplets and those that were put in table 1, however, why did not specify the size of droplets in the table 1, if the scale of the size of the droplets was specified in the material and methods section?  

>The size of the Droplet in this experiment is about the size that it dropped on the inkjet paper and smeared. It is not added to Table 1 because it is necessary to measure the size of the flying Droplet precisely. Droplet size measurement is not sufficient in this experiment. The discussion also describes the limitations of this study.

Page 10, Line 280-283

However, this evaluation system does not directly measure the droplet size and cannot detect floating droplets that do not fall. Therefore, future studies on aerosol visualization technology, such as droplet image velocimetry (PIV), are required.

Line 293 to 294. “The use of an…” Would it be possible to describe how many toothpaste is recommended according to their results?  

> I can't mention the specific amount of dentifrice because it varies from person to person such as saliva, but if it is 1.0g or more, it is thought that the droplets can be suppressed. It seems that the maximum amount that can be placed on the bristles of a toothbrush is about 1.5g.

Lines 283 to 287. "Our results..." This paragraph is interesting, however, is confusing in the lines 293 to 294 "The use of an..." in which the amount of toothpaste is described, would it be possible to review this paragraph and redact it in better manner? 

> The reviewer's comment is correct. We have deleted the “without requiring a change in the amount of toothpaste”

Page10, Line306-311

Our results indicate that, by reducing the dilution of the toothpaste, the concentration of fluoride and the above-mentioned antiviral components could be kept high, without re-quiring a change in the amount of toothpaste. It is recommended to use more than 1.0 g of fluoride-containing toothpaste for caries prevention, but it may be preferable to use a larger amount to prevent droplet generation [19].

Reviewer 4 Report

Comments to Authors

            The authors have presented a manuscript entitled “Relationship between toothpaste dilution ratio and droplets generated during toothbrushing”, referring to an experimental study on droplets generation during toothbrushing, regarding toothpaste dilution.

            I congratulate the authors on their study; however, I have some concerns on the manuscript in its present form which I’ll leave below for consideration.

            The Reviewer

Major Reviews

            - In the introduction section, authors keep revisiting the covid-19 pandemic to support interest on the present work. However, please acknowledge that oral hygiene procedures are still prohibited in most of situations, or at least they are not encouraged. Same as for everything that requires the removal of the mask. I suggest that authors rewrite introduction to give more value to the present experimental study on a day-by-day basis, and not give such importance to the pandemic. Maybe leave a minor note like “in the covid-19 pandemic context, this is also of high importance to consider when addressing public health….”. Please consider. Also, toothpaste is not supposed to be diluted before use. Authors should clarify it and turn the text around to give importance on the mixing of toothpaste and saliva which in turn may generate droplets by dilution. Please consider.

            - In the Materials and Methods section, authors state the groups division. However, in the results, more groups are shown. Please explain in the first section all the groups considered in analysis. Also, along the results section, most/all figures/graphs/tables do not present the data for all the groups. This lack of information may give the impression that authors are selecting preferred data that is in accordance to the hypothesis at study. (ex. Group 1.50x is never shown along the results).

            - Authors need to clarify how was the developed method validated. This is, how sure are the authors that the developed method is a good way to assess the parameters evaluated, and there is a need to justify the p value considered significant and why was 5 independent measures considered “enough”.

Minor Reviews

- page 1/lines 40-44: ok, this is true. However, these components are not maintained for long periods of time in the oral cavity (depending on brushing frequency).

- along the materials and methods section there is a lot of brand/manufacturer, city and country data missing (example: toothbrush head, inkjet paper, components of toothpaste, and so on), please revise.

- page 2/line 76-79: how do authors support this design? Any published similar protocol?

- page 2/line 81: add reference

- page 2/line 81-83: why changing the toothbrush head design? How reproductible in terms of usual commercially available toothbrushes is the one used?

- page 2/line 85: were the repeated test independent or not? Please clarify.

- page 2/line 87-88: what was recorded and analyzed, the mean of the 5 repeated measures?

- Figure 1: the edge does not replicate the palatal side of teeth and the movement towards incisal edge will never have such kinetic as the one in the design.

- Why is the first dilution 67% with 1.5x and bot 75%? Also, nomenclature should be revised in terms of groups division. It is not easily related the x’s with the %. First time reading I got confused if 1.00x received or not dilution.

- page 4/line 125: this n may lead into consufion.

- page 4/line 125-128: was normal distribution verified before using parametric tests?

- page 4/line 136: what do authors mean by “most marked”, is it the highest mean difference?

- page 4/line 139-140: correlation coefficient was not mentioned in statistical analysis nor is here.

- Figure 2 – why does group x1.00 corresponds to 100% water content? Please revise.

- section 3.2 to 3.6: authors have to describe results by giving values and the exact yielded p value. Please revise all results. Also, repeated information should be avoided.

- section 3.2 – the first time that the foam group shows up without being explained. Please revise by adding criteria for this group back in the materials and methods section.

- All figures: revise writing, meaning of symbols goes into the caption of the respective figure. Caption of figure 4 also contains statistical analysis, please revise.

- page 8/line 194: redundant and inaccurate: if no droplets were produced, the flight distance does not equal zero. It just does not exist. Please revise.

- Figure 5: There are droplet counts in distances above the maximum flight distance described before. Please revise.

- Page 8/line 214: I do not agree. It was not suppressed but reduced.

- Section 6: this analysis does not seem to add any interesting data. Can authors increase its interest for the manuscript?

Author Response

<Comments to Authors>

The authors have presented a manuscript entitled “Relationship between toothpaste dilution ratio and droplets generated during toothbrushing”, referring to an experimental study on droplets generation during toothbrushing, regarding toothpaste dilution.

I congratulate the authors on their study; however, I have some concerns on the manuscript in its present form which I’ll leave below for consideration.

>Thank you very much for providing important comments. We are thankful for the time and energy you expended. Our responses to the referees’ comments are as follow:

<Major Reviews>

- In the introduction section, authors keep revisiting the covid-19 pandemic to support interest on the present work. However, please acknowledge that oral hygiene procedures are still prohibited in most of situations, or at least they are not encouraged. Same as for everything that requires the removal of the mask. I suggest that authors rewrite introduction to give more value to the present experimental study on a day-by-day basis, and not give such importance to the pandemic. Maybe leave a minor note like “in the covid-19 pandemic context, this is also of high importance to consider when addressing public health….”. Please consider. Also, toothpaste is not supposed to be diluted before use. Authors should clarify it and turn the text around to give importance on the mixing of toothpaste and saliva which in turn may generate droplets by dilution. Please consider.

>We appreciate the reviewer's comment on this point.

As you commented, oral hygiene procedures are still prohibited in most of situations, or at least they are not encouraged. We conducted this study to change this situation. We think toothbrushes are important for maintaining and improving oral health and should not be regulated. In this study, it was found that oral care can coexist with infection prevention if dentifrice is used properly. The dentifrice is mixed with saliva in the oral cavity and diluted. However, if the amount of dentifrice used is sufficient, it will not be diluted, and the number of droplets can be reduced. As a result, we believe that it will also prevent infection.

> In accordance with the reviewer's comment, we have added the following sentence:

Page2, Line 57-58

In the covid-19 pandemic context, this is also of high importance to consider when ad-dressing public health.

Page1, Line11-15 (ABSTRACT)

Guidelines for using toothpaste during tooth-brushing in public places during the coronavirus epidemic are lacking. In addition, the advantages and disadvantages of using toothpaste in terms of droplet generation during brushing, the number of droplets generated, and their scatter range are unknown. Therefore, we investigated the relationships of diluted toothpaste viscosity, the number of droplets generated, and the droplets’ flight distance.

- In the Materials and Methods section, authors state the groups division. However, in the results, more groups are shown. Please explain in the first section all the groups considered in analysis. Also, along the results section, most/all figures/graphs/tables do not present the data for all the groups. This lack of information may give the impression that authors are selecting preferred data that is in accordance to the hypothesis at study. (ex. Group 1.50x is never shown along the results).

> The reviewer's comment is correct. We appreciate the reviewer's comment on this point. We have added following sentence to Method section.

Page3, Line116-119

The toothpaste was diluted with distilled water to prepare five groups, with concentra-tions of 100% (1.00×), 67% (1.50×), 50% (2.00×), 25% (4.00×), and 10% (10.0×). 4.00× foams well, and in the foam state, the behavior at the time of droplet generation is different. Therefore, it was divided as 4.00× (Form). Additionally, water without toothpaste was used for comparison.

There is a reason why the groups shown in the figures and tables are different.

This study is broadly divided into two parts.

 1. Experiment to clarify the relationship between viscosity and dilution ratio (Fig. 2, Table 1)

 2. Experiment to clarify the relationship between the generation / distance of droplets and the dilution  

  ratio (Fig. 3-6, Table 1)

Experiment 1 has 1.50x data, but 2 does not collect 1.50x experimental data.

Therefore, Figure 3-6 is unified with water, 1.0x, 2.0x, 4.0x, 10x.

- Authors need to clarify how was the developed method validated. This is, how sure are the authors that the developed method is a good way to assess the parameters evaluated, and there is a need to justify the p value considered significant and why was 5 independent measures considered “enough”.

> This experiment will be the first trial. In-vivo verification will be in the future, but we think it is important to have a sufficient range that can be expected in the oral cavity from no dilution to 10.0x dilution. We think it is more important that the range of condition settings and a certain degree of relevance were found, rather than five.

<Minor Reviews>

- page 1/lines 40-44: ok, this is true. However, these components are not maintained for long periods of time in the oral cavity (depending on brushing frequency).

> The reviewer's comment is correct. We also think that the brushing frequency is a factor that affects the effect.

- along the materials and methods section there is a lot of brand/manufacturer, city and country data missing (example: toothbrush head, inkjet paper, components of toothpaste, and so on), please revise.

>We appreciate the reviewer's comment on this point. We regret this mistake.

We added brand/manufacturer, city and country data in Method section.

Page 2, Line85-86

The structure of the toothbrush head (Original prototype for this research (not for sale), KAO Corp., Tokyo, Japan) was simplified.

Page 2, Line88-90

The droplets created by the robot were dropped onto inkjet paper (KA4100SFR, Super Fine paper, A4, Seiko Epson Corp.,Tokyo, Japan, measurement range: W210 × L594 mm, connect two A4 sheets) and recorded.

- page 2/line 76-79: how do authors support this design? Any published similar protocol?

> There is no reference as this is the original method we developed.

However, a similar experiment has been posted on IJERPH for electric toothbrushes.

  1. Erwin P. Mark, Michael A. O. Lewis, Filippo Graziani, Boris Atlas,* and Joern Utsch

Droplet Sizes Emitted from Demonstration Electric Toothbrushes

Int J Environ Res Public Health. 2021 Mar; 18(5): 2320.

Published online 2021 Feb 26. doi: 10.3390/ijerph18052320

- page 2/line 81: add reference

> There is no reference as this is the original method we developed. This paper will be the first publication.

- page 2/line 81-83: why changing the toothbrush head design? How reproductible in terms of usual commercially available toothbrushes is the one used?

> We aimed to clarify and simplify the factors involved in the formation of droplets.

By making only one row of tufts, the influence of hair transplanted there (thickness, shape, material, etc.) can also be clarified. We believe that using commercial products is not suitable for basic research in the early stages, as many factors may be involved.

In a preliminary study, we tested nylon hair with 3 levels of thickness (6, 8, 10 mil) and 7.5 mil super-tapered PBT hair with the same design. From our experiment, the effect of viscosity was the largest on the generation of droplets. By simplifying the head, the contribution of each element can be clarified.

We would like to report the comparison with the commercial products in future research. We haven't tried many, but even if we use a commercially available toothbrush, it is reproduced that the higher the viscosity, the less the droplets are generated.

We believe that this paper is also useful in the sense that the contribution of each element can be clarified when examining commercial products and in-vivo in the future.

- page 2/line 85: were the repeated test independent or not? Please clarify.

- page 2/line 87-88: what was recorded and analyzed, the mean of the 5 repeated measures?

> We appreciate the reviewer's comment on this point.

The repeated test was independent. We analyzed the mean of the 5 repeated measures.

We have added following sentence to this section.

Page 2, Line 90-92

The measurement sample was printed on an inkjet by generating splashes by reciprocating motion 5 times. Measurements were performed 5 times for each dilution. The inkjet paper was changed after each measurement.

- Figure 1: the edge does not replicate the palatal side of teeth and the movement towards incisal edge will never have such kinetic as the one in the design.

> Since this research is a model system, the shape of the teeth is not accurately reproduced. Toothbrush bristles bend and are released to generate droplets. This model is a reproduction of the phenomenon that gives a certain amount of deflection to the toothbrush when splashes occur and releases it.

- Why is the first dilution 67% with 1.5x and bot 75%? Also, nomenclature should be revised in terms of groups division. It is not easily related the x’s with the %. First time reading I got confused if 1.00x received or not dilution.

> We appreciate the reviewer's comment on this point.

Since the concentration is wt %, we think it will be related. The 1.50x bot in Figure 2 is 67%. Please confirm.

To avoid confusion, we have added “no dilution” following sentence to Method section.

Page 3, Line 116-119

The toothpaste was diluted with distilled water to prepare five groups, with concentrations of 100% (1.00×, no dilution), 67% (1.50×), 50% (2.00×), 25% (4.00×), and 10% (10.0×). 4.00× foams well, and in the foam state, the behavior at the time of droplet generation is different.

- page 4/line 125: this n may lead into consufion.

> The reviewer's comment is correct. We have deleted the “n=5”.

- page 4/line 125-128: was normal distribution verified before using parametric tests?

> The Shapiro-Wilk test has verified that nonparametric tests can be used. (Origin Pro 2022, OriginLab Corp. USA,Massachusetts).

- page 4/line 136: what do authors mean by “most marked”, is it the highest mean difference?

> We agree with your assessment. It means the difference between the mean values.

we have changed sentence.

Page 4, Line 147-149

The amount of decrease of 100% (1.00 ×) and 67% (1.50 ×) was the largest; after 1.5×, the decrease became more gradual. 

- page 4/line 139-140: correlation coefficient was not mentioned in statistical analysis nor is here.

> We regret this mistake. We added scatter plot and correlation coefficient sentence in Method section.

Page 4, Line 140-143

A scatter plot was created to clarify the relationship between the concentration and viscosity of toothpaste, the number of droplets generated. The correlation coefficient was calculated from the approximate expression.

- Figure 2 – why does group x1.00 corresponds to 100% water content? Please revise.

> We think that dentifrice that is not diluted with water can be described as 100%. We consider that percentage by weight (wt%) of a target substance in a homogenous material. Concentration of A = Weight of A (g) / Overall weight including water (g)

- section 3.2 to 3.6: authors have to describe results by giving values and the exact yielded p value. Please revise all results. Also, repeated information should be avoided.

> In accordance with the reviewer's comment, we have added P-value to following sentence:

Page 5, Line 182-183

Water and 1.00× samples showed statistically significant differences from all other dilu-tions (p < 0.01) (Fig. 4a).

- section 3.2 – the first time that the foam group shows up without being explained. Please revise by adding criteria for this group back in the materials and methods section.

> In accordance with the reviewer's comment, we have added following sentence in Method section:

Page 3, Line 116-119

The toothpaste was diluted with distilled water to prepare five groups, with concentrations of 100% (1.00×), 67% (1.50×), 50% (2.00×), 25% (4.00×), and 10% (10.0×). 4.00× foams well, and in the foam state, the behavior at the time of droplet generation is different. Therefore, it was divided as 4.00× (Form).

- All figures: revise writing, meaning of symbols goes into the caption of the respective figure. Caption of figure 4 also contains statistical analysis, please revise.

>In accordance with the reviewer's comment, we have added meaning of symbols goes into the caption

Page 5, Line 162-165

The viscosity and correlation of each toothpaste concentration (1.00×, 1.50×, 2.00×, 4.00× and 10.0×) are shown. The white square indicates dentifrice with a concentration of 100% (1.00×). White cir-cles; 67% (1.50×), white triangles; 50% (2.00×), Black inverted triangles; 25% (4.00×), Gray diamonds; 10% (10.0×).

Page 7, Line 205-207

The black inverted triangles indicated dentifrice with a water. White triangles; 10% (10.0×), White circles; 25% (4.00×), white square; 50% (2.00×), Gray diamonds; 100% (1.00×).

Page 8, Line 230-232

The black square indicated dentifrice with a water. White circles; 10% (10.0×), black triangles; 25% (4.00×), white inverted triangles; 25% (4.00×, form), black diamonds; 50% (2.00×).

We have deleted the description of statistical analysis in Figure 4.

- page 8/line 194: redundant and inaccurate: if no droplets were produced, the flight distance does not equal zero. It just does not exist. Please revise.

>The reviewer's comment is correct. We have deleted the ” (0 mm flight distance)”.

Page 8, Line 216-217

In the case of 1.00×, no droplets were observed (Fig. 4c).

- Figure 5: There are droplet counts in distances above the maximum flight distance described before. Please revise.

>In accordance with the reviewer's comment, we have revised figure 5.

- Page 8/line 214: I do not agree. It was not suppressed but reduced.

>In accordance with the reviewer's comment, we have changed this to following sentence:

Page 8, Line 238-240

Comparing 4.00× with and without foaming, the number of droplets generated was reduced when foaming was present.

- Section 6: this analysis does not seem to add any interesting data. Can authors increase its interest for the manuscript?

> We think that showing the moment of splash generation with an image deepens the reader's understanding. Especially, the difference between 4.00x and 4.00x (Form) is difficult to convey only by sentences, so we think that an image is necessary.
